# Bioaccessibility and Bioavailability of (-)-Epigallocatechin Gallate in the Bread Matrix with Glycemic Reduction

**DOI:** 10.3390/foods12010030

**Published:** 2022-12-21

**Authors:** Lanqi Li, Jing Gao, Hui Si Audrey Koh, Weibiao Zhou

**Affiliations:** Department of Food Science and Technology, National University of Singapore, Science Drive 2, Singapore 117542, Singapore

**Keywords:** EGCG, glycemic reduction, bioaccessibility and bioavailability, starch digestion in vitro, molecular docking

## Abstract

Bread has a high glycemic index (GI) and rich contents of quickly digestible carbohydrates, which is associated with insulin resistance and the risk of chronic diseases. (-)-Epigallocatechin Gallate (EGCG) is the primary catechin component that inhibits starch hydrolases, while the low release and absorption rates limit its utilization. In this study, EGCG was added to the bread matrix for fortification to reduce its glycemic index compared to white bread. EGCG fortification at 4% decreased the starch digestion rate of baked bread by 24.43% compared to unfortified bread and by 14.31% compared to white bread, with an identical amount of EGCG outside the matrix. Moreover, the predicted GI (pGI) was reduced by 13.17% compared to white bread. Further, 4% EGCG-matched bread enhanced the bioaccessibility and bioavailability of EGCG by 40.38% and 47.11%, respectively, compared to the control. The results of molecular docking demonstrated that EGCG had a higher binding affinity with α-amylase than with α-glucosidase, indicating that EGCG may effectively inhibit the accumulation of carbs during starch digestion. Thus, EGCG can be used as a functional ingredient in bread to reduce its glycemic potential, and the bread matrix can be used as a carrier for EGCG delivery to enhance its bioaccessibility and bioavailability.

## 1. Introduction

Bread is a typical staple food consumed in many nations that is high in carbohydrates and contains a lot of quickly digestible starch, which frequently results in a high glycemic index (GI). The GI value describes the contribution of food products to blood glucose response in the body based on the carbohydrate quality, but not the quantity of the food. To better evaluate the overall glycemic effect of foods, a concept, namely glycemic load (GL), was introduced as the product of GI and the carbohydrate content in a typical serving. Epidemiologic evidence strongly supports the excessive consumption of high dietary GI and GL food was associated with insulin resistance and the risk of some chronic diseases, such as cardiovascular diseases and type 2 diabetes mellitus [1]. Therefore, developing a functional bread with low GI as a potential meal replacement for sick people would somewhat ameliorate the deterioration of chronic diseases.

The dietary intervention of phytochemicals, such as flavonoids, terpenoids, and polysaccharides, has been proven to be an effective treatment to reduce the activities of digestive starch enzymes, therefore, causing a reduction in GI and GL and improving the phytochemical function through synergistic interaction with the food matrix [2]. Bread is a semi-solid oil–water mixed system containing protein, lipids, carbohydrates, and other bioactive molecules. Phytochemicals would synergize with bread matrices and promote their bioactivity compared with pure phytochemicals [3]. Several studies have reported fortified bread, such as quercetin-fortified bread, fucoidan-fortified bread, and anthocyanin-fortified bread [4,5,6]. These types of bread have been proven to reduce the bread GI effectively.

Among numerous natural phytochemicals, (-)-epigallocatechin gallate (EGCG), the principal component in green tea catechins, has the most significant capacity to inhibit α-glucosidase and α-amylase [7,8]. In recent studies, EGCG has also been applied to glucose control and body fat metabolism and, therefore, EGCG might act as a promising compound in preventing diabetes [9,10]. However, the utilization of bioactive function is limited by its poor bioaccessibility, bioavailability, absorption, distribution, and metabolism [11]. The bioactive molecules delivered through the food matrix system might be more functional than pure substances. Bread is an ideal food-delivering carrier for bioactive natural products, while EGCG is the key phytochemical for the retarding of carbohydrate digestion. Bread with EGCG may be a practical approach for creating low-GI food products [12].

There has been lots of work reporting on tea-catechin-fortified bread, assessing its quality attributes, sensory evaluation, catechin stability, antioxidant properties, and health benefits, indicating that EGCG-fortified bread is a potential functional food for anti-diabetes [4,12,13]. However, the mechanism behind the phenomenon is still unknown. In this study, the related research will go a step further. The comparative matrix effects between pure phytochemical supplements and phytochemicals incorporated into the bread matrix will be discussed, indicating the possibility of EGCG-fortified bread as a potential meal replacement for diabetes. 

This study aimed to investigate the starch digestive profile and matrix effects of EGCG-fortified bread compared with the matched one (control bread mixed with EGCG, equivalent amount to the corresponding fortified bread, outside the food matrix). Moreover, the potential of utilizing the bread matrix to enhance the bioavailability and bioaccessibility of EGCG was evaluated through a simulated digestion model. The effects of EGCG on bread starch digestion, GI, and GL values were investigated. In addition, the potential molecular interactions of EGCG with starch hydrolase were analyzed to explore the mechanism of starch inhibition.

## 2. Materials and Methods

### 2.1. Materials

EGCG (Teavigo, food grade, EGCG > 94%, caffeine < 1%) was obtained from (Taiyo Kagaku Co. Ltd., Yokkaichi, Japan). The ingredients were obtained from the local Fair Price supermarket: high-protein wheat bread flour (Prima, 13.1% protein), instant active dry yeast (*Saccharomyces cerevisiae*, S.I.Lesaffre, Maisons-Alfort Cedex, France), refined salt, pure cane sugar, and shortening. For the chemistry assay, chemicals were purchased from Sigma-Aldrich Co., Ltd. (St. Louis, MO, USA): Methanol, HCl, NaOH, 3,5-Dinitrosalicylic acid (DNS).

### 2.2. The Making of Bread Samples

The EGCG-fortified bread-making protocol was adopted from previous research. Bread dough was prepared by mixing 200 g wheat bread flour after adding EGCG at 1, 2, and 4%, respectively (the dosage was based on Wang’s research on the bread properties and sensory evaluation [14]). In a mixer, 120 g water, 6 g shortening, 6 g sugar, 2 g instant dry active yeast, and 2 g salt are mixed first slowly for 1 min and then rapidly for 5 min (KPM50, KitchenAid, Benton Harbor, MI, USA). After resting for 10 min at 20 °C, the dough was shaped into small pieces (55 g each) and proofed in a proofer (KBF115, Binder, Tuttlingen, Germany) at 40 °C and 85% RH. We then spent 8 min baking at 200 °C in an oven (MS01T04-2, Eurofours, La Longueville, France). After cooling the cooked bread sample to room temperature for 1 h, the crust and crumb were separated dependently, and only the crumb was collected for further experiment. After that, the crumb was mixed and filtered with a 40-micron sieve to obtain bread powders [5].

### 2.3. EGCG Extraction and Detection

To measure the retention rate of EGCG in the bread before digestion, 5 g lyophilized bread-crumb powders was blended with 25 mL of methanol in 50 mL beakers using an orbital shaker (IKA VXR basic Vibrax, Staufen Co. Ltd., Köngen, Germany) at 600 rpm for 2 h in a 60 °C water bath. Following this, liquid extract and solid fraction were obtained by centrifugation at 3500× *g* (Eppendorf 5810R, Thermo Fisher Scientific Co. Ltd., Waltham, MA, USA) for 5 min. Five-times the solid fraction was re-extracted using the process described previously. For High-Performance Liquid Chromatography (HPLC) analysis, the mixed liquid extract was diluted to 50 mL with methanol.

### 2.4. Quantitative Detection of EGCG Using HPLC/DAD

The quantification of EGCG utilized a HPLC equipped with a diode array detector (DAD) (Shimadzu, Kyoto, Japan) and a C18 reserved-phase column (250 × 4.6 mm, 5 µm, 100 Å, Sunfire, Waters, Wexford, Ireland). The detection protocols were adapted from Wang’s research: Mobile phase A (1% acetic acid in DI water) and B (100% methanol) were applied according to the gradient elution with 1 mL/min flow rate, and 30 °C column oven temperature at 275 nm wavelength detection. Quantitative detection of EGCG calibration was accomplished by the external calibration ranging from 0.025 to 0.250 mg/mL [15]. The calibration curve of EGCG is shown in Appendix A.

### 2.5. Simulated Digestion In Vitro and Dialysis of Bread Samples

The simulated digestion in vitro protocols were adopted from the latest standardized static INFOGEST 2.0 [16]. The schematic flowchart of in vitro simulated digestion and dialysis of bread samples is shown in Figure 1. Simulated Salivary Fluid (SSF), Simulated Gastric Fluid (SGF), and Simulated Intestinal Fluid (SIF) were all prepared using an identical methodology. Thus, 5 g bread-crumb aliquots was combined with 5 mL SSF buffer during the oral phase (1:1 wt./v, including 1.5 mM CaCl_2_, 75 U/mL α-amylase, pH 7.0). After vortexing the suspension for 20 s, it was then incubated for 2 min at 37 °C in circulating water baths (MX-CA21E, Polyscience, Niles, IL, USA) with a magnetic stirring rod on a magnetic stirrer (MIXdrive 15, 2mag AG, Muenchen, Germany). In the gastric phase, oral bolus was mixed with 8 mL SGF buffer (1:1 vol/vol, including 2000 U/mL pepsin, 60 U/mL gastric lipase, 0.15 mM CaCl_2_, pH 3.0). After that, the mixture was blended using a stirring rod at 37 °C for 2 h. In the intestinal phase, gastric chyme was blended with SIF buffer (1:1 vol/vol, 10 mM bile, 0.6 M CaCl_2_, 100 U/mL trypsin, pH 7.0). The resultant mixture was then transferred into a 14 kDa cut-off-size dialysis tube, submerged in 200 mL of phosphate-buffered saline (PBS, pH 7.0), and dialyzed for 6 h at 37 °C [17]. For digestion rate modeling, at certain time intervals of up to 5 h, dialysate aliquots were taken in a 1.5 mL centrifuge tube for further detection.

### 2.6. Detection of Released Reducing Sugars

In the intestinal phase, a 0.5 mL aliquot of dialysate was collected from 0 to 180 min at 5 min intervals. The released reducing sugar in the dialysate at each time point was determined by using the DNS approach [18]. We combined 0.5 mL of DNS reagent with 0.5 mL of dialysate samples (1:1 vol/vol) for 10 min in boiling water. After being cooled to room temperature, the absorbance of the combination was measured at 540 nm [19].

### 2.7. Calculation of the EGCG Bioaccessibility and Bioavailability

After the intestinal phase, the digesta in the dialysis tube were centrifuged at 18,000× *g* for 10 min, and the supernatant was collected and brought to 50 mL in a flask with DI water. In the interim, the dialysate was first concentrated to less than 50 mL at 40 °C using a vacuum evaporator (N-1200A, Eyela Co. Ltd., Tokyo, Japan) and then brought to 50 mL in the volumetric flask using DI water. The digesta were transferred into a 50 mL centrifuge tube and centrifuged. After that, the EGCG quantification by HPLC was held to measure its amount in the digesta and dialysate [20].

The bioaccessibility and bioavailability were calculated by using Equations (1) and (2), respectively.
(1)$$F_{AC}\;\left(\%\right)=\frac{A_{dig}+A_{dia}}{A_{i}}\times 100\%$$
(2)$$F_{AV}\;\left(\%\right)=\frac{A_{dia}}{A_{i}}\times 100\%$$

*A_dig_* is the contents of EGCG detected in the digesta, *A_dia_* is the contents of EGCG in the dialysate, and *A_i_* is the contents of EGCG in the bread after baking. ‘Dialysate’ refers to the solution outside the dialysis tube and ‘digesta’ refers to the solution in the dialysis tube [6].

### 2.8. EGCG Recovery Rate from Digesta

For fortified and matched samples, the recovery of EGCG from the digesta was determined using Equation (3)
(3)$$R_{rec}=\frac{A_{dig}}{A_{ini}}\times 100\%$$
where *R_rec_* is the recovery rate. *A_ini_* is the total quantity of EGCG in bread aliquots after baking and before digestion, whereas *A_dig_* is the total contents of EGCG detectable in the digesta.

### 2.9. Mathematical Simulation of the Starch Digestion Kinetics

The starch digestion kinetics of bread aliquots followed a thermodynamic first-order reaction, which can be modeled using Equation (4) to determine the digestive profile of the bread aliquots.
(4)$$C_{t}=C_{\infty }\left(1-e^{-kt}\right)\;$$
where *C_t_* is the concentration of released reducing sugar (mg/mL) over time *t* (min) and $C_{\infty }$ is the equilibrium concentration of reducing sugar; *k* is the rate constant of starch digestion (min^−1^). Non-linear regression was performed to achieve the *k* and $C_{\infty }$ values. In addition to *R^2^*, the root mean square error (RMSE) was employed to evaluate the quality of the established model [21].

### 2.10. Assessment of Total Available Carbohydrates (TAC)

The total accessible carbohydrate contents in the bread aliquots were determined using an assay kit from Megazyme (K-ACHDF 08/16, Megazyme Co. Ltd., Wicklow, Ireland) [22]. The bread crumb was suspended in MES-Tris buffer using a high-speed stirrer (Ultra Turrax T25, Janke and Kunkel IKA-Labortechnik, Staufen, Germany) following the protocol provided by Megazyme [19].

### 2.11. Calculation of pGI and pGL

Assessment of the in vitro simulated digestion was conducted to calculate the predicted Glycemic Index (*pGI*) and predicted Glycemic Load (*pGL*) based on Equations (5)–(8) developed by Goñi [23]. The curve of reducing sugar concentration (mg/mL) released over time (min) from Section 2.6 was expressed in terms of grams of reducing sugar released per 100 g of TAC of fresh bread determined in Section 2.10. For each sample, the area under the hydrolysis curve (*AUC*) up to 180 min was integrated. By dividing the AUC of each bread sample by the reference, the control white baked bread, the hydrolysis index (*HI*) was computed (Equation (7)). White bread was chosen as the reference food for *GI* (*GI bread*, white bread = 100) (Equation (5)), while glucose served as the reference for estimating *GI glucose* (*GI glucose*, glucose = 100) (Equation (6)). Each sample’s generated *GI bread* was then multiplied by 0.7. Based on a 50 g portion of bread and taking into account the *TAC* content of each sample, the predicted *GL* was determined (Equation (8)) [24].
(5)$$pGI_{bread}=0.549HI+39.71$$
(6)$$pGI_{glucose}=pGI_{bread}\times 0.7$$
(7)$$HI=\frac{AUC_{sample}}{AUC_{control\;wheat\;bread}}\times 100$$
(8)$$pGL=\frac{\mathrm{pGI}\times \mathrm{TAC}}{100}$$

### 2.12. Computational Molecular Docking

Computational molecular docking was employed to explore the potential binding of EGCG with α-amylase and α-glucosidase using AutoDock Vina 1.2.3 to illustrate the mechanism that EGCG inhibits the enzyme activities [25]. The α-amylase and α-glucosidase structures were obtained from the RCSB Protein Data Bank (PDB ID:6z8l, 7kry, respectively). Chem3D Ultra 12.0 was used to draw the three-dimensional structure of EGCG, whereas MM2 reduced the energy [26]. The docking data with the lowest docking affinity were determined as the most appropriate result. PyMol 2.3 was used to visualize and generate the docking results.

### 2.13. Statistical Analysis

All the experiments were conducted in triplicates. Results were presented as mean ± standard deviation (SD). One-way analysis of variance (ANOVA) with Duncan was used to determine the significant differences in the results. The non-linear curve fitting, regression, and integration were performed using OriginPro software (2021, OriginLab Corporation, Northampton, MA, USA) and all statistical analysis was conducted using SPSS 27.0 (IBM Co. Ltd., Chicago, IL, USA) [21].

## 3. Results and Discussion

### 3.1. Bread Starch In Vitro Digestibility

To examine the influence of EGCG in the bread matrix on starch digestion and absorption, control bread aliquots were supplemented with the same quantity of EGCG as remained in the fortified bread samples after digestion (denoted as EGCG-matched bread, set as the positive control against the fortified one). Figure 2 depicts a dose-dependent inhibitory connection between EGCG and the contents of released reducing sugar (RS) during in vitro starch digestion of fortified and matched bread aliquots.

The curves continually rise from 0 to 100 min and level off afterward. At the end of dialysis, the released RS concentration was significantly decreased due to the formation of phenolic-starch complexes. It has been reported that the released reducing sugar of bread enriched with EGCG significantly decreased during pancreatic digestion [13]. In this study, at the end of the starch digestion (t = 180 min), the contents of reducing sugar (RS) in the 4% EGCG-fortified bread samples were 24.43%, significantly lower than the concentration in the control bread samples, while the reduction rate of RS concentration in the 4% EGCG-matched bread was significant at 11.81%. It suggested that EGCG dose-dependently inhibits starch digestion in bread samples, as the contents of released RS decrease with increasing EGCG concentration.

With increasing EGCG content in the bread matrix, the inhibitory effects of bread starch digestion increased, according to these findings. The inhibitory impact of EGCG on α-amylase was one probable explanation. As one of the principal phenolics, EGCG inhibits starch digestion by pancreatin, α-amylase, α-glucosidase, and other digestive enzymes to a greater extent [27]. It is likely attributable to the fact that the hydroxyl groups of the A and B rings of EGCG engage noncovalently with the enzyme binding sites. The development of EGCG–enzyme complexes weaken the hydrophobic contact and hydrogen bonding between the -OH groups of the EGCG structure and the catalytic residues of the active binding pocket in α-amylase [28].

### 3.2. Mathematical Modeling of Starch Digestion Curves

Based on the mathematical modeling of the digestion profile, the equilibrium concentration *C_∞_* of RS and the digestion rate constant *k* value were estimated (Equation (1)). As demonstrated in Table 1, the *k* value decreased from 0% to 4% EGCG concentration in both fortified and unfortified bread. A similar pattern applied to the value of *C_∞_*. The slower the rate of in vitro starch digestion, the lower the *k* value would be. Sui et al., showed a 20.5% decrease in the regression digestion coefficient of the 4% black rice extract anthocyanin-enhanced bread compared to the control bread in their investigation [4].

The RS concentrations of EGCG-fortified bread aliquots were significantly lower than the control, and the parameters of the fitting curves could quantitatively discriminate the digestion profiles further. Compared to the control bread samples, the contents of released reducing sugar at the end of starch digestion were 1.27%, 8.50%, and 15.70% lower in the EGCG-fortified bread aliquots at 1%, 2%, and 4%; meanwhile, the starch digestion rate *k* was 21.76%, 20.42%, and 24.43% higher than control bread with significance. It could be attributed to the fact that EGCG could interact with starch digestive and hydrolyze enzymes through hydrogen bonding and hydrophobic interactions, reducing the contents of rapidly digested starch in bread and inhibiting the starch enzyme digestion activities [2].

### 3.3. Calculation of pGI and pGL

The summarization of *pGI* and *pGL* for bread aliquots was listed in Table 2. A dose-dependent reduction in *AUC*, *HI*, *pGIbread*, *pGIglucose*, and *pGL* was observed for all the EGCG-fortified bread aliquots compared to the control. There were significant differences in *pGI* and *pGL* in the fortified with matched bread at a relatively higher EGCG dosage (4% and 2%), while no significant differences were observed at the lower dosage (1%). The maximum reduction in *pGIbread* in the fortified/matched bread samples was observed at 13.17% and 8.38%, respectively. Meanwhile, the maximum reduction in *pGL* value in the fortified/matched bread was 20.69% and 16.35%, respectively. Both the *pGIbread* and *pGL* value of fortified bread were significantly lower than that of the matched bread. It might be attributed to the formation of resistant starch. In the fortified bread, EGCG could interact with starch in the noncovalent binding, contributing to the generation of resistant starch, which was able to reduce the *pGI*, while in the matched bread, EGCG interacts with bread starch loosely out of the bread delivery system, forming a relatively unstable force with starch.

Moreover, starch digestion was hindered by the glutelin, gliadin, and lipids in the bread matrix. EGCG–protein complexes were proven to be formed in wheat bread delivery systems with the addition of black tea [29], which might make protein more challenging to digest by the protease. As a consequence, the starch is less likely to be exposed to the environment of the amylase hindering by the protein. In addition, EGCG could interact with the enzyme in the digestive system and weaken the enzyme activity to some extent [30].

### 3.4. Bioaccessibility and Bioavailability of EGCG after Digestion

The terms ‘bioaccessibility’ and ‘bioavailability’ are used to evaluate the utilization efficacy of nutrients. ‘bioavailability’ refers to the nutrient release rate from the food matrix, while ‘bioavailability’ refers to the nutrient absorbed rate by the small intestine epithelium [31].

The percentage of EGCG preserved in bread crumbs after baking is presented in Table 3. After baking at 200 °C, the retention rate of EGCG in bread was considerably lower. This could be attributable to the part of EGCG that cannot be extracted due to interactions between EGCG and wheat protein. The capacity to extract EGCG from the bread matrix was impacted by developing EGCG–protein complexes via hydrophobic interactions and hydrogen bonding.

On the other hand, the degradation and epimerization of EGCG would also be contributing factors. As reported by Mizukami et al., the number of cis-catechins decreased continually under the roasting at 180 °C, due to the degradation and epimerization [32]. Compared with other baked food-delivery systems, such as cookie and cake, bread was proven to be a better carrier for EGCG fortification. EGCG would undergo degradation and epimerization in the different food-delivery systems. Zhang et al. reported an 88% loss of EGCG in cookies after 10 min baking at 200 °C, resulting from the EGCG degradation and epimerization in a lipid-rich alkaline food-delivery system [33].

Both bioaccessibility and bioavailability are essential indexes for the evaluation of available components from specific food matrices. In vitro bioaccessibility refers to the contents of EGCG released after digestion in the liquid fraction of the digesta and dialysis as a fraction of the EGCG that was initially in the bread after baking. As passive diffusion was one of the main methods of EGCG absorption, the in vivo bioavailability refers to the EGCG released after digestion in the dialysate passing through the dialysis tube as a fraction of the EGCG that was initially retained in the bread after baking. Significant differences are observed between the fortified and the matched bread at the different dosages of EGCG. There was lower bioaccessibility of EGCG (32.34%) in the 4% matched bread compared with the fortified one (45.40%) (Table 3). A similar result can also be found in bioavailability (%) and other dosages. This phenomenon indicated the synergistic interaction between the bread matrix and EGCG, enhancing the absorption of potential EGCG. EGCG could interact with gluten/gliadin in the bread during baking via hydrogen bonding and hydrophobic interactions. This finding also suggested that close mixing of EGCG with bread ingredients resulted in homogeneous dispersion of EGCG in the bread matrix, which provided a significant effective surface area for poorly water-soluble compounds to be available for dissolution and may have helped to increase the bioavailability of compounds.

### 3.5. Computational Simulation of EGCG on α-Amylase and α-Glucosidase

It was reported that EGCG could strongly inhibit the activity of both α-amylase and α-glucosidase, but the molecular mechanism has still not been fully elucidated, especially under the circumstance of food matrix [34]. Both α-amylase and α-glucosidase are the key enzymes in starch digestion. In this study, the interaction of EGCG on α-amylase and α-glucosidase was identified and verified using the method of molecular docking. The result of molecular docking with the lowest binding affinity is shown in Figure 3. The binding affinity of EGCG with α-amylase was −9.5 kcal/mol, while the binding affinity of EGCG with α-glucosidase was −8.4 kcal/mol. It meant that EGCG was prone to binding with α-amylase and forming a tighter complex. This indicated that EGCG was more selective to α-amylase than α-glucosidase and would not lead to the accumulation of carbohydrates during starch digestion, thus, effectively avoiding the flatulence phenomenon caused by the fermentation of carbohydrates by anaerobic bacteria in the large intestine [35].

The binding pocket of α- amylase with EGCG included Trp 59, Tyr 62, Gln 63, His 101, Gly 104, Leu 162, Thr 163, Leu 165, Arg 195, Asp 197, Ala 198, Lys 200, His 201, Glu 233, Val 234, Ile 235, Asp 300, and Gly 306 with 18 amino acid residues. The binding pocket of α-glucosidase with EGCG included Phe 307, Ala 427, Ala 429, Asp 456, Arg 459, Phe 461, Thr 462, Pro 465, Thr 466, Arg 467, Pro 469, Val 495, Asp 496, Gly 498, Tyr 499, Arg 500, Val 501, Trp 523, Cys 524, Trp 525, Met 546, Met 565, and Phe 571 with 23 amino acid residues. The binding pocket is relatively more minor in the α-glucosidase than α- amylase, resulting in the more flexible binding of EGCG with α-amylase, which meant EGCG could more easily enter the binding site and form a tight complex.

## 4. Conclusions

In this study, simulated starch digestion in vitro was performed to assess the bread starch digestion kinetics, the *pGI* and *pGL* value of bread with EGCG incorporation, as well as the bioaccessibility and bioavailability of EGCG. Compared to the control bread samples, the content of reducing sugar released at the end of starch digestion was 1.27%, 8.50%, and 15.70% lower in the EGCG-fortified bread samples at 1%, 2%, and 4%, while the starch digestion rate was 21.76%, 20.42%, and 24.43% higher than control bread with significance. Further, 4% EGCG-fortified bread significantly reduced the *pGIglucose* and *pGL* by 13.17% and 20.69%, respectively, compared to the control. The resultant *pGIglucose* of the 4% EGCG-fortified bread at 60.78 ± 1.29 was lower than most high-carbohydrate foods, proven to be a healthy food with low glycemic potential for consumers. The matrix effect of bread on the starch digestive enzyme inhibition activity of EGCG was investigated. The 4% EGCG-fortified bread possessed a significantly lower starch digestion rate than that of the 4% matched EGCG bread. One possible mechanism is that the fortification of EGCG into bread before baking allowed EGCG to interact with starch via hydrophobic interactions and hydrogen bonding to change the starch structure of bread samples, therefore, hindering the digestion of starch, which is more efficient compared with the interaction in the solution or outside the bread matrix. Furthermore, incorporating 4% EGCG into the bread matrix could increase the bioaccessibility and bioavailability by 45.39% and 55.60%, respectively. One possible explanation is that EGCG could conjugate with gluten and form the disulfide bond through the SH-SS interchange reaction during the dough mixing and avoid being unstable throughout the digestion process and, therefore, increase the small intestinal absorption. The result reveals that the incorporation of EGCG into bread systems could significantly reduce the starch digestion rate of the bread samples and lower their *pGI* and *pGL* value compared with the EGCG out of the bread-delivery system. The molecular docking results effectively explain the binding affinity of EGCG with α-amylase and α-glucosidase, the disrupted structure of α-amylase and α-glucosidase with the insertion of EGCG in the active binding sites avoiding the entrance of substrate and decrease the enzyme activity ultimately. Therefore, bread was proven to be a promising food carrier for delivering EGCG more efficiently. The results of this study highlighted the feasibility of EGCG being used as a functional ingredient in the reformulation of bread for its glycemic reduction.

## Figures and Tables

**Figure 1 foods-12-00030-f001:**
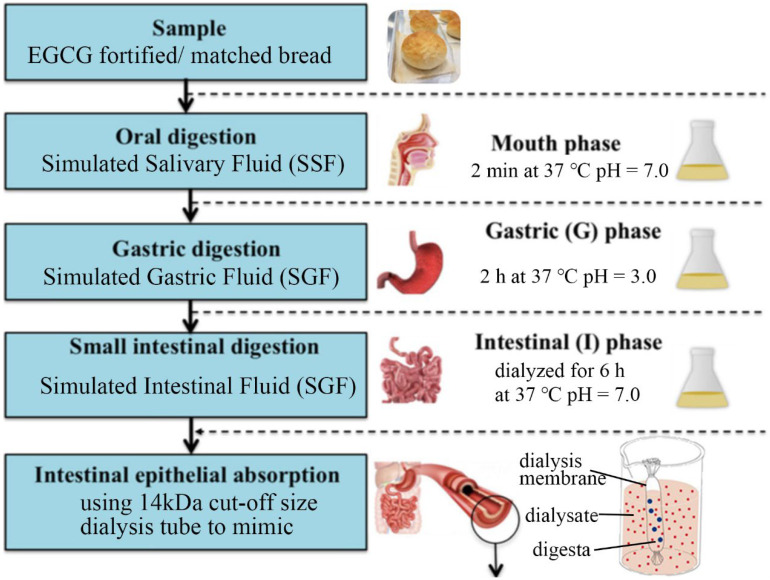
A schematic flowchart of in vitro simulated digestion and dialysis.

**Figure 2 foods-12-00030-f002:**
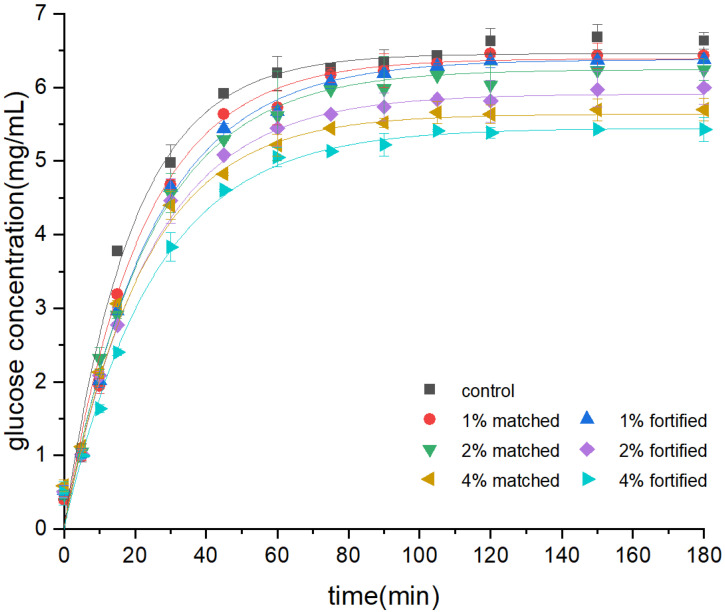
The concentration of reducing sugar released over time for in vitro digestion kinetics of 0, 1, 2, and 4% EGCG fortified bread and matched bread (control bread mixed with EGCG equivalent to fortified bread) and the corresponding fitting curves.

**Figure 3 foods-12-00030-f003:**
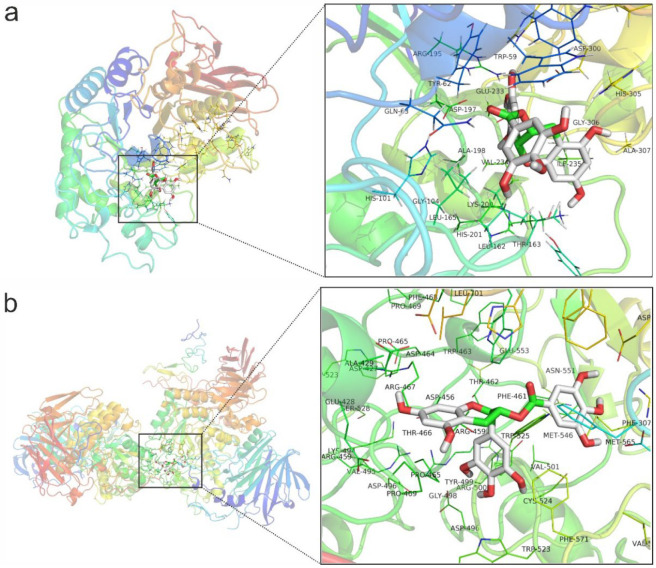
The interactions of EGCG with α-amylase (**a**) and α-glucosidase (**b**) computationally simulated by AutoDock Vina with the lowest docking affinity. α-amylase and α-glucosidase are shown in cartoon form. The detailed view showed the adjacent residue of the binding site in the active pocket, labeled with amino acid and the corresponding sequence number.

**Table 1 foods-12-00030-t001:** Modeling parameters of in vitro digestion curve.

Model	BoxLucas1		Equation	$C_{t}=C_{\infty }\left(1-e^{-kt}\right)$			
Plot	Control	1% Matched	1% Fortified	2% Matched	2% Fortified	4% Matched	4% Fortified
$C_{\infty }$	6.4610 ± 0.0080 ^a^	6.3926 ± 0.0208 ^b^	6.3798 ± 0.0066 ^b^	6.2450 ± 0.0031 ^b^	5.9121 ± 0.0075 ^c^	5.6387 ± 0.0292 ^c^	5.4466 ± 0.0069 ^d^
k	0.0524 ± 0.0027 ^a^	0.0459 ± 0.0013 ^b^	0.0410 ± 0.0007 ^b^	0.0419 ± 0.0003 ^b^	0.0417 ± 0.0007 ^c^	0.0449 ± 0.0017 ^c^	0.0396 ± 0.0009 ^d^
Reduced Chi^2^	2.3182	1.6722	1.3716	0.4497	0.7962	1.4566	3.2006
R^2^	0.9996	0.9997	0.9998	0.9999	0.9998	0.9997	0.9996
RMSE	1.5225	1.2931	1.1711	0.6706	0.8923	1.2069	1.7890

^a,b,c,d^ Numbers with different upper/lowercase letters are significantly different in the same row (*p* < 0.05).

**Table 2 foods-12-00030-t002:** The glycemic reduction characteristics in vitro of bread fortified and matched with EGCG.

	Control	1% Matched	1% Fortified	2% Matched	2% Fortified	4% Matched	4% Fortified
*AUC*/g min 100 g^−1^	1039.68 ± 13.23 ^a^	1011.54 ± 15.08 ^b^	992.87 ± 16.89 ^b^	974.98 ± 16.55 ^b^	922.58 ± 16.60 ^c^	889.46 ± 15.43 ^c^	842.97 ± 17.48 ^d^
*HI*/%	100.00 ± 1.27 ^a^	97.29 ± 1.45 ^b^	95.50 ± 1.62 ^b^	93.78 ± 1.59 ^b^	88.74 ± 1.60 ^c^	85.55 ± 1.48 ^c^	81.08 ± 1.68 ^d^
*pGI bread*/%	100.00 ± 1.27 ^a^	98.43 ± 1.47 ^b^	97.39 ± 1.66 ^b^	96.39 ± 1.64 ^b^	93.46 ± 1.68 ^c^	91.62 ± 1.59 ^c^	86.83 ± 1.85 ^d^
*pGI glucose*/%	70.00 ± 0.89 ^a^	68.90 ± 1.03 ^b^	68.17 ± 1.16 ^b^	67.47 ± 1.15 ^b^	65.42 ± 1.18 ^c^	64.13 ± 1.11 ^c^	60.78 ± 1.29 ^d^
*pGL*/%	17.74 ± 0.23 ^a^	17.12 ± 0.26 ^b^	16.56 ± 0.28 ^b^	15.96 ± 0.27 ^b^	15.14 ± 0.27 ^c^	14.84 ± 0.26 ^c^	14.07 ± 0.29 ^d^

*AUC* = area under the curve; *HI* = hydrolysis index; *pGI bread* = predicted glycemic index relative to control bread; *pGI glucose* = predicted glycemic index relative to glucose; *pGL* = predicted glycemic load. Values are the means ± SEM, *n* = 6 per treatment. ^a,b,c,d^ Within the same row, means lacking a common alphabetic letter differ (*p* < 0.05).

**Table 3 foods-12-00030-t003:** Quantification of EGCG in the dialysate and digesta of fortified/matched bread at different EGCG dosages after in vitro digestion.

Bread Type	EGCG Dosage	Retention Level of EGCG before the In Vitro Digestion/%	EGCG in the Dialysate/mg per 5 g Bread	EGCG in the Digestate/mg per 5 g Bread	Bioaccessibility/%	Bioavailability/%
fortified	1%	70.068 ± 0.459 ^c^	2.947 ± 0.002 ^c^	6.469 ± 0.040 ^c^	26.877 ± 1.725 ^c^	8.412 ± 0.068 ^c^
	2%	75.496 ± 0.378 ^b^	8.311 ± 0.025 ^b^	18.075 ± 0.028 ^b^	34.949 ± 0.740 ^b^	11.008 ± 0.651 ^b^
	4%	79.314 ± 1.213 ^a^	22.715 ± 0.631 ^a^	49.306 ± 1.296 ^a^	45.402 ± 5.344 ^a^	14.320 ± 2.602 ^a^
matched	1%	N/A	2.703 ± 0.018 ^c^	6.540 ± 0.141 ^c^	18.486 ± 2.812 ^d^	5.406 ± 0.367 ^d^
	2%	N/A	8.997 ± 0.003 ^b^	17.880 ± 0.173 ^b^	26.877 ± 1.725 ^c^	8.997 ± 0.031 ^c^
	4%	N/A	19.467 ± 0.107 ^a^	45.218 ± 0.736 ^a^	32.342 ± 3.681 ^b^	9.734 ± 0.535 ^c^

N/A refers to not applicable. ^a,b,c,d^ Numbers with different upper/lowercase letters are significantly different within the same row (*p* < 0.05).

## Data Availability

Data is contained within the article or Appendix A.

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
