# Peer review of "Bioaccessibility and Bioavailability of (-)-Epigallocatechin Gallate in the Bread Matrix with Glycemic Reduction"

_foods, 2022, doi:10.3390/foods12010030_

Round 1

Reviewer 1 Report

The manuscript entitled “In vitro Bioaccessibility and Bioavailability of (-)-Epigallocate- 2 chin Gallate in the Bread Matrix with Glycemic Reduction”, it’s a work dealing with the fortification of breads with EGCG to reduce its glycemic index compared to white bread. The work has a very interesting topic, which can have great impact in bread functionalization, where EGCG fortification can be used as a functional ingredient in bread to reduce its glycemic potential 

Material and methods

Line 112 – The calibration curve of EGCG should be added.

Line 120 – change wt/wt to wt/v.

Author Response

Thanks for your comments!

Line 112 – The calibration curve of EGCG should be added.

Ans: I have attached the calibration curve in Supplementary data (Figure S1), please see the attachment.

Line 120 – change wt/wt to wt/v.

Ans: I have changed the corresponding term

✅ 启动解除复制限制 ✅ 启动解除右键限制 ✅ 启动解除键盘限制          

Reviewer 2 Report

Dear authors, excellent work, I accept the manuscript in its present format.

Author Response

Thanks for your comments!

✅ 启动解除复制限制 ✅ 启动解除右键限制 ✅ 启动解除键盘限制          

Reviewer 3 Report

This study investigated the in vitro bioaccessibility and bioavailability of (-)-Epigallocatechin Gallate in the bread matrix with glycemic reduction. The manuscript is well-planned, and the results are interesting. Although there were some similar papers, this work reported new aspects of this field. 

For all tables, please use the letter "a" for the largest value.

Author Response

Thanks for your comments!

For all tables, please use the letter "a" for the largest value.

Ans: I have changed the corresponding terms for all the tables.